# Internalized Sexual Stigma among Lesbian, Gay, and Bisexual Individuals in Taiwan: Its Related Factors and Association with Mental Health Problems

**DOI:** 10.3390/ijerph19042427

**Published:** 2022-02-19

**Authors:** Jia-In Lee, Yu-Ping Chang, Ching-Shu Tsai, Cheng-Fang Yen

**Affiliations:** 1Department of Psychiatry, Kaohsiung Medical University Hospital, Kaohsiung 80756, Taiwan; 1050644@kmuh.org.tw; 2Department of Psychiatry, School of Medicine, Graduate Institute of Medicine, College of Medicine, Kaohsiung Medical University, Kaohsiung 80708, Taiwan; 3School of Nursing, The State University of New York, University at Buffalo, Buffalo, NY 14260, USA; yc73@buffalo.edu; 4Department of Child and Adolescent Psychiatry, Chang Gung Memorial Hospital, Kaohsiung Medical Center, Kaohsiung 83301, Taiwan; 5School of Medicine, Chang Gung University, Taoyuan 33302, Taiwan; 6College of Professional Studies, National Pingtung University of Science and Technology, Pingtung 91201, Taiwan

**Keywords:** psychological well-being, sexual minority, stigma, sexual minority

## Abstract

Internalized sexual stigma (ISS) is one of the major issues that can compromise the health of sexual minority populations. This quantitative study aimed to examine: (1) the relationships of individual factors (gender, age, education level, sexual orientation, and age of identification of sexual orientation) and perceived family support with ISS; and (2) the associations of ISS with mood problems and the moderating effects of gender on the associations among Taiwanese young adult lesbian, gay, and bisexual (LGB) individuals. In total, 500 male and 500 female young adult LGB individuals aged between 20 and 30 years participated in this study. The experience of ISS, individual (e.g., gender, age, education level, sexual orientation, and age of identification of sexual orientation) and environmental factors (perceived family), mood problems (e.g., anxiety and depression) were collected. The individual and environmental factors related to ISS and the associations of ISS with mood problems were examined using multivariate linear regression analysis. The results indicated that gender, sexual orientation, age of identification of sexual orientation, and perceived family support were significantly associated with all or some dimensions of ISS in LGB individuals. Various dimensions of ISS had different relationships with anxiety and depression. Gender had moderating effects on the association between the identity dimension of ISS and sexual orientation as well as between the social discomfort dimension of ISS and anxiety. Various dimensions of ISS among LGB individuals should be routinely assessed by mental health service units. Intervention programs should be provided for LGB individuals, especially those with factors related to ISS.

## 1. Introduction

### 1.1. Internalized Stigma

Internalized stigma is a process whereby individuals endorse stereotypes about their personal characteristics, such as race and ethnicity, health status, body shape, and gender or sexual orientation. Individuals with internalized stigma may anticipate social rejection, consider stereotypes to be self-relevant, and believe that they are devalued members of society [1,2]. Internalized stigma is prevalent among individuals with mental illnesses [3], HIV-AIDS [4], obesity [5], and drug and alcohol use [6] as well as among racial or ethnic [7] and sexual and gender minorities [8]. Studies have revealed that internalized stigma affects individuals’ self-esteem, medical help-seeking behavior, and health outcomes [3,4,5,6,7,8]; therefore, internalized stigma should be examined in depth and suitable interventions should be identified.

### 1.2. Internalized Sexual Stigma in Lesbian, Gay, and Bisexual Individuals

Lesbian, gay, and bisexual (LGB) individuals experience multiple types of social stigma, such as structural stigma (discrimination and stigmatization at the institutional and sociocultural levels e.g., the ban on same-sex marriage) and bullying and hate crimes derived from heterosexism [9], and they may develop internalized sexual stigma (ISS) [10]. Research has revealed that ISS is a multifactorial construct [11,12,13]. According to the Measure of Internalized Sexual Stigma for Lesbians and Gay Men (MISS-LG) [14], ISS is composed of three fundamental dimensions: identity, social discomfort, and sexuality. The identity dimension corresponds to an enduring propensity to have a negative self-attitude as a member of a sexual minority and to consider sexual stigma as part of a value system and identity (“I’d prefer to be heterosexual”; “If it were possible, I’d do anything to change my sexual orientation”); social discomfort is the fear of public identification as a lesbian or gay man in social contexts and the fear of disclosure in private and professional life (e.g., “At university (and/or at work), I pretend to be heterosexual”; “It’s difficult for me to say that I’m lesbian/gay, even to someone I know”); and the sexuality dimension describes the pessimistic evaluation of the quality and duration of intimate gay or lesbian relationships and a negative conception of gay or lesbian sexual behaviors (e.g., “I don’t believe in love between homosexuals”; “Gay men can only have flings/one-night stands”) [14]. Health professionals should consider all dimensions of ISS when developing prevention and intervention programs for reducing ISS in LGB individuals.

### 1.3. ISS in LGB Individuals during Early Adulthood

Research has demonstrated that ISS may compromise LGB individuals’ mental health [15,16,17,18] and social relationships [19], increase the risk of unprotected sexual behaviors [17,20,21], and decrease the intention to access medical care services [22]. ISS may also mediate the relationship between enacted stigma and depressive symptoms in young men who have sex with men (MSM) [23]. As in other Asian communities, unfavorable attitudes and unfriendly behaviors toward LGB individuals in Taiwan are deep rooted in the mindset of many people and are difficult to change [24,25,26,27,28,29,30]. Given that young adulthood is a phase of life in which individuals pursue independence and explore life possibilities [31], in this period, ISS may hinder the development of self-identity, social interaction, and intimate relationships in LGB individuals. Identifying the individual and environmental factors related to ISS in LGB individuals during early adulthood can highlight factors that should receive emphasis in efforts to better educate the broader society regarding their views on LGB individuals as well as within the context of psychotherapy in the event that an LGB individual feels the need for such therapy.

### 1.4. Rationale for This Study

Several factors regarding ISS warrant further study. First, studies have considered ISS to be a single construct [32,33,34,35,36,37]. However, according to the MISS-LG [14], ISS is composed of three dimensions. Whether the factors related to ISS vary across the distinct dimensions warrants further study. Second, research has identified several sociodemographic and environmental factors that are related to ISS. For example, gender [14], age [14,37], sexual orientation [35,36], social support [37], information on sexual minorities [37], and perceived social stigma [32,33,34,36,38] are significantly associated with ISS in LGB individuals. The manner in which ISS is experienced by lesbians and gay men is considerably different [38]; however, whether gender moderates the relationships between ISS and individual and environmental factors has not been examined. Third, ISS is positively associated with delayed acceptance of sexual orientation among young LGB individuals [39]; however, whether age of identification of sexual orientation is significantly associated with ISS is underdetermined. Because of the significant association between early identification of sexual orientation and bullying victimization in LGB individuals [40], we hypothesized that LGB individuals who identify sexual orientation at a younger age have higher ISS compared with those who identify sexual orientation at an older age.

### 1.5. Aims

This cross-sectional survey study in young adult LGB individuals in Taiwan had three aims. First, we examined the factors, including demographics, sexual orientation characteristics, and perceived family support, related to ISS. Second, we examined the relationships between ISS and anxiety and depression. Third, we examined the moderating effects of gender on the associations of ISS with the related factors of anxiety and depression. We hypothesized that: (1) the levels of ISS would be different among young adult LGB individuals with different demographics, sexual orientation characteristics, and perceived family support; (2) higher ISS would be associated with higher anxiety and depression; and (3) gender would moderate the relationships between ISS and the related factors of anxiety and depression.

## 2. Methods

### 2.1. Participants and Procedure

The Investigation on Stigma among LGB Individuals in Taiwan recruited 1000 participants (500 men and 500 women). The methods used to enroll the participants and some of the study results have been described elsewhere [41,42]. In brief, we posted an advertisement on social media sites frequently used by Taiwanese LGB individuals, such as Facebook, Twitter, and the LINE messaging app, and on the Bulletin Board System and the home pages of three health promotion centers for LGB individuals between August 2018 and July 2020. The inclusion criteria were individuals aged between 20 and 30 years who identified as homosexual or bisexual and lived in Taiwan. Those who met the inclusion criteria were invited to complete the study questionnaires individually in a study room. Those who had any condition that might interfere with their understanding of the study’s purpose or how to complete the questionnaire, such as impaired intellect or alcohol and substance use, were excluded from the study. Informed consent was obtained from all participants prior to assessment. Participants completed the research questionnaires in the study rooms of the psychiatry research department affiliated to a university hospital in person and being assured that their responses to the research questionnaire would be confidential. This study provided 500 new Taiwan dollars (about 18 US dollars) for every participant as a reward. This study was approved by the Institutional Review Board of Kaohsiung Medical University Hospital (KMUHIRB-F(II)-20180018).

### 2.2. Measures

#### 2.2.1. ISS

We used the 17-item Chinese version [42] of the MISS-LG [14] to assess the three ISS dimensions of social discomfort, sexuality, and identity in LGB individuals. The items were rated on a 5-point Likert scale from 1 (strongly disagree) to 5 (strongly agree). Some of the sample items for each dimension have been described in the introduction of this paper. The MISS-LG has two versions with the same factor structure, one for women and the other for men. A higher total dimension score indicates a higher level of ISS. According to research, the MISS-LG has satisfactory psychometric properties [14]. The results of Rasch and confirmatory factor analyses verified that the traditional Chinese version of the MISS-LG (TC-MISS-LG) has the same three-factor structure for the different genders used among young adult LGB individuals in Taiwan [42]. The TC-MISS-LG scores were significantly correlated with perceived social stigma toward sexual minorities, supporting its concurrent validity; the McDonald’s omega of the three TC-MISS-LG dimensions ranged from 0.67 to 0.90, supporting its acceptable-to-excellent internal consistency [42].

#### 2.2.2. Perceived Family Support

We used the 5-item traditional Chinese version [43] of the Family APGAR Index [44] to measure the five components of family support: adaptability, partnership, growth, affection, and resolve. The items were rated on a 4-point Likert scale from 1 (never) to 4 (always). A higher total score indicated a higher level of perceived family support. The traditional Chinese version of the Family APGAR Index had acceptable discriminatory validity for social adaptability [43] and congruent validity, exhibiting a significant correlation with the state of general health [45]. Cronbach’s α in the present study was 0.86.

#### 2.2.3. Demographic and Sexual Orientation Factors

We collected information on the participants’ gender, age, education level (high school or below vs. college or above), sexual orientation (homosexual or bisexual), and age of identification of sexual orientation.

#### 2.2.4. Anxiety

We used the 20-item traditional Chinese version [46] of the self-administered State-Trait Anxiety Inventory-State subscale [47] (TC-STAI-S) to assess participants’ current severity of anxiety. The items were rated on a 4-point Likert scale from 1 (not at all) to 4 (very much so). A higher total TC-STAI-S score indicated more severe anxiety. The traditional Chinese version of the TC-STAI-S had acceptable test–retest reliability (Pearson’s r = 0.76), internal reliability (Cronbach’s α = 0.91), criterion validity (correlation with the Hamilton Anxiety Rating Scale: r = 0.69), and construct validity [48]. Cronbach’s α for the TC-STAI-S in the present study was 0.89. The score of 40 is commonly used to define probable clinical levels of anxiety [49].

#### 2.2.5. Depression

We used the 20-item traditional Chinese version [50] of the self-administered Center for Epidemiological Studies Depression Scale [51] (TC-CES-D) to assess the frequency of depressive symptoms among participants in the month preceding the study. The items were graded on a 4-point scale from 0 (rarely or none of the time) to 3 (most or all of the time). A higher total TC-CES-D score indicated more severe depression. The TC-CES-D had good internal consistency (Cronbach’s α = 0.90), 1-week test–retest reliability (intraclass correlation reliability = 0.93), congruent validity (area under the receiver operative characteristic curves for major depressive disorder = 0.88–0.90) [52], and construct validity [53]. Cronbach’s α for the TC-CES-D in the present study was 0.93. The score of 16 is commonly used for screening major depressive disorder in the general population [54].

### 2.3. Statistical Analysis

Data analyses were conducted using IBM SPSS software (version 20.0; IBM, Armonk, NY, USA). Participants’ ISS, individual factors (demographic and sexual orientation characteristics), perceived family support, mood problems (anxiety and depression) were analyzed using descriptive statistical methods (mean, standard deviation [SD] and percentage). We first examined the skewness and kurtosis of the continuous variables and found that all the absolute values were less than 2, indicating normal distributions according to Kim [55]. The associations between individual factors and perceived family support (independent variables) and the three ISS dimensions (dependent variables) were examined using multivariate linear regression analysis. Because of multiple comparisons, the significance of the *p* value was adjusted to 0.017 (0.05/3) using Bonferroni correction. The associations between ISS (independent variable) and anxiety and depression (dependent variables) were also examined using multivariate linear regression analysis, which was controlled for demographics (covariates). Because of multiple comparisons, the significance of the *p* value was adjusted to 0.025 (0.05/2) using Bonferroni correction. The moderating effects of gender on the associations of ISS with the related factors and with anxiety and depression were examined according to the method of Baron and Kenny [56].

## 3. Results

Table 1 summarizes the data on the three ISS dimensions, demographics, sexual orientation factors, perceived family support, anxiety, and depression of the 1000 participants. The mean (SD) age of participants was 24.6 years (3.0 years); nearly 90% had a college degree or above; 57% identified as homosexual; and the mean (SD) age at which they first identified their sexual orientation was 14.5 (3.9) years. The mean total scores (SD) for the dimensions of social discomfort, sexuality, and identity, based on the MISS-LG, were 16.6 (6.0), 8.9 (3.3), and 9.9 (4.2), respectively. After transforming, the mean score for each item on the dimension of social discomfort (2.4) was the highest, followed by the scores for identity (2.0) and sexuality (1.8). The mean score (SD) for the severity of anxiety was 40.8 (12.7) and slightly higher than the cutoff 40 that was commonly used for screening anxiety disorders [49]. The mean score (SD) for depression was 18.8 (11.2) and higher than the cutoff 16 that was commonly used for screening major depressive disorders [54].

Table 2 presents the results of the multivariate linear regression analysis of the associations between demographics, sexual orientation factors, and perceived family support and the three ISS dimensions. The condition index was 29.981, indicating no problem of collinearity. The results indicated that male participants had higher ISS in all three dimensions than female participants. Participants identifying as homosexual had lower ISS in all three dimensions than those identifying as bisexual. An older age of identification of sexual orientation was identified as being significantly associated with higher ISS in the dimensions of social discomfort and sexuality. Lower perceived family support was significantly associated with higher ISS in the dimension of social discomfort.

Table 3 presents the results of the multivariate linear regression analysis of gender’s moderating effects on the association between the three ISS dimensions and related factors. The results indicated that the interaction between gender and sexual orientation was significantly associated with the identity dimension of ISS, indicating that gender moderated the association between this dimension and sexual orientation. Further examination revealed that bisexual men had higher identity-related ISS than gay men (B = −2.321, SE = 0.470, *p* < 0.001), whereas no difference in identity-related ISS was observed between bisexual and lesbian women (B = −0.415, SE = 0.366, *p* = 0.257).

Table 4 provides the results of the multivariate linear regression analysis of the associations between ISS and anxiety and depression. The condition index was 29.929, indicating no problem of collinearity. Higher ISS in the dimension of social discomfort was significantly associated with higher anxiety and depression, higher ISS in the dimension of sexuality was significantly associated with higher anxiety, and higher ISS in the dimension of identity was significantly associated with higher depression.

Table 5 provides the results of the multivariate linear regression analysis of gender’s moderating effects on the associations between ISS and anxiety and depression. The results indicated that the interaction between gender and the social discomfort dimension of ISS was significantly associated with anxiety, indicating that gender moderated the association between social discomfort and anxiety. Further examination revealed that social discomfort was significantly associated with anxiety in lesbian and bisexual women (B = 0.708, SE = 0.127, *p* < 0.001), but this was not the case for gay and bisexual men (B = 0.152, SE = 0.133, *p* = 0.252).

## 4. Discussion

The present study demonstrated that gender, sexual orientation, age of identification of sexual orientation, and perceived family support were significantly associated with all or some dimensions of ISS in LGB individuals. Various dimensions of ISS had different relationships with anxiety and depression. Gender had a moderating effect on the association between the identity dimension of ISS and sexual orientation as well as the on association between the social discomfort dimension of ISS and anxiety.

### 4.1. Factors Related to ISS

The MISS-LG has two versions, one for men and the other for women, with the same three first-order factors, the same number of items, and some different item contents related to gender [14]. It provides a good basis for comparing ISS-related gender differences. In line with the results of previous studies [14,57,58,59], the results of the current study revealed that gay and bisexual men reported higher ISS in all three dimensions than lesbian and bisexual women. Gay and bisexual men endure greater pressure to conform to a heteronormative gender role and are more often condemned by their heterosexual peers [14,60,61]. Gay and bisexual men may experience more public stigma based on heterosexualism and thus comprehensively internalize sexual stigma more than do lesbian and bisexual women.

The present study revealed that bisexual men and women had higher ISS in the dimensions of social discomfort and sexuality than gay men and lesbians. The results are congruent with those of previous studies [37,62,63,64]. A 1-year longitudinal study also demonstrated that bisexual youth experienced ISS more than did gay or lesbian youth [65]. The present study further revealed an association between the ISS dimension of identity and the sexual orientation of bisexuality in men, but not in women. Research has demonstrated that bisexual individuals experience prejudice, for example sexual orientation instability and sexual irresponsibility, from both heterosexuals and lesbians and gay men [66,67,68], which may increase the risk of developing ISS as part of a value system and in relation to identity [69]. The result indicated that the effect of social prejudice on internalizing stigma toward the sexual orientation of bisexuality might be stronger in men than in women.

A previous study in the United States identified that peer support, but not family support, was associated with low ISS among young MSM aged between 16 and 20 years [37]; the authors hypothesized that at this stage of development, peer support plays a larger role than family support in determining the experience of ISS [37]. However, the present study discovered that lower perceived family support was significantly associated with higher ISS in the dimension of social discomfort among young adult LGB individuals in Taiwan. Most young adults in Taiwan maintain a close relationship with their family and are concerned about the expectations and attitudes of their elders toward them. People in Taiwan are deeply influenced by Confucianism; adults who are unmarried and have no offspring are considered to have failed in observing filial piety [70,71,72]. Young adult LGB individuals may internalize their failure to meet their family obligations and develop ISS, especially with regard to the fear of disclosure in family life.

The present study determined that a younger age of identifying sexual orientation was associated with higher ISS in the dimensions of social discomfort and sexuality. Because early adolescence is a stage for exploring and developing self-identification, LGB individuals who identify their sexual orientation at this stage may lack the ability to reject social stigma toward sexual minorities. They may also lack the LGB approbation that may enable them to develop positive self-identification regarding sexual orientation. These developmental perspectives may contribute to ISS development among LGB individuals with early identification of sexual orientation.

### 4.2. Associations of ISS with Anxiety and Depression

The present study revealed that higher ISS in the social discomfort dimension was significantly associated with higher anxiety and depression. The relationship between ISS and mood problems might be bidirectional. According to Erikson [73], young adults are eager to develop intimacy with others. Young adult LGB individuals who experience social discomfort-related ISS may have difficulties in developing intimate relationships with others, and the risk of anxiety and depression may increase. Moreover, establishing sexual orientation and initiating relationships are key developmental tasks during adolescence [74]. The ISS dimensions of sexuality and identity may be formed in early adolescence and may interfere with the accomplishment of developmental tasks at the end of adolescence; the adverse impact may persist into early adulthood and compromise emotional regulation. Alternatively, mood problems might increase LGB individuals’ sensitivity to the existence of sexual stigma and might increase the severity of ISS. The present study also revealed that the social discomfort dimension of ISS was significantly associated with anxiety in lesbian and bisexual women, but not in gay and bisexual men. Studies have established that women are more likely than men to have anxiety disorders [75,76,77]; however, the present study could not determine the reasons for the moderating effect of gender. Further research is required to examine this factor.

### 4.3. Implications

The findings of the present study highlight the value of developing strategies for the prevention of ISS in LGB individuals. While acceptance of LGB individuals has substantially improved over the past several decades, total acceptance is far from complete due to the continued prevalence of generally unwarranted negative familial and cultural stereotypes and thus further efforts to modify outdated beliefs with regard to LGB individuals are warranted. Constitutions, laws and anti-discrimination policies at the national level that include protection from discrimination on the grounds of sexual orientation is of fundamental importance [78]. Broadening understanding of LGB culture and awareness of prejudices toward LGB individuals in educational settings, workplaces, and family environments are also the necessary steps to help reduce sexuality-related stigma [78,79]. Public health strategies addressing attitudes to sexual orientation and promoting the changes of attitudes toward sexual minority among the general population may contribute to diverse affirmative cultural scripts regarding LGB individuals’ lives [78,80]. Although several studies have developed the programs for reducing ISS among gay and bisexual men [81,82,83], there is no study directly examining the effects of interventions addressing familial and cultural stigma on ISS among LGB individuals. Meanwhile, given the effect of gender on the associations between ISS and sexual orientation and anxiety identified in this study, intervention programs should be gender-specific.

### 4.4. Limitations

Some limitations of this study should be considered when interpreting the results of this study. First, the temporal relationships among ISS, family support, and mood problems could not be determined in this cross-sectional study. Further prospective studies are needed to examine the temporal relationships among these variables; for example, whether ISS can predict the development or change of mood problems warrants being examined in prospective studies. Second, the present study recruited a group of LGB individuals aged between 20 and 30 years. Therefore, further study is needed to examine whether the results of this study can be generalized to the populations of other age ranges. Third, we collected the data based on the participants’ self-report; there might be single-rater bias. Fourth, the options for the question inquiring participants’ gender identity contained man and woman only but no other gender identities such as transgender, gender nonbinary, and genderqueer.

## 5. Conclusions

Gender, sexual orientation, age of identification of sexual orientation, and perceived family support were related to the ISS dimensions in young adult LGB individuals. ISS was significantly associated with anxiety and depression. Further efforts to modify the public’s prejudice with regard to LGB individuals are warranted. Family education and cultural change aiming to reduce public stigma against LGB individuals are needed to mitigate ISS among LGB individuals.

## Figures and Tables

**Table 1 ijerph-19-02427-t001:** Participants’ characteristics (*N* = 1000).

	*n* (%)	Mean (SD)	Range
Gender			
Female	500 (50)		
Male	500 (50)		
Age (years)		24.6 (3.0)	20–30
Education level			
High school or below	109 (10.9)		
College or above	891 (89.1)		
Sexual orientation			
Bisexual	430 (43)		
Homosexual	570 (57)		
Age of identification of sexual orientation (years)		14.5 (3.9)	5–29
Perceived family support (5 items)		136 (3.6)	5–20
Internalized sexual stigma on the MISS-LG		35.3 (11.5)	17–76
Social discomfort (7 items)		16.6 (6.0)	7–34
Sexuality (5 items)		8.9 (3.3)	5–22
Identity (5 items)		9.9 (4.2)	5–23
Anxiety (20 items)		40.8 (12.7)	20–79
Depression (20 items)		18.8 (11.2)	0–57

MISS-LG: Measure of Internalized Sexual Stigma for Lesbians and Gay Men.

**Table 2 ijerph-19-02427-t002:** Factors related to internalized sexual stigma: Multivariate linear regression analysis.

	Social Discomfort	Sexuality	Identity
	B (SE)	B (SE)	B (SE)
Gender	**4.105 (0.371) *****	**4.500 (0.165) *****	**2.121 (0.270) *****
Age	0.112 (0.061)	0.005 (0.027)	0.017 (0.044)
Education	0.928 (0.572)	−0.060 (0.254)	0.150 (0.416)
Sexual orientation	**−2.206 (0.409) *****	**−1.115 (0.182) *****	**−1.249 (0.298) *****
Age of identification of sexual orientation	**0.132 (0.049) ****	**0.066 (0.022) *****	0.031 (0.036)
Perceived family support	**−0.184 (0.049) *****	−0.035 (0.022)	−0.050 (0.036)

** *p* < 0.01; *** *p* < 0.001. **Bold**: Indicates significance using Bonferroni correction, which adjusted the *p* value to 0.017.

**Table 3 ijerph-19-02427-t003:** Moderating effects of gender on the associations of internalized sexual stigma with related factors: Multivariate linear regression analysis.

	Social Discomfort	Sexuality	Identity
	B (SE)	B (SE)	B (SE)
Gender	4.568 (2.147)	**4.438 (0.754) *****	**3.562 (0.415) *****
Age	0.111 (0.061)	0.004 (0.027)	0.014 (0.044)
Education	0.927 (0.573)	−0.059 (0.254)	0.134 (0.412)
Sexual orientation	**−1.926 (0.554) ****	**−0.868 (0.246) *****	−0.132 (0.384)
Age of identification of sexual orientation	0.128 (0.070)	0.054 (0.031)	0.034 (0.036)
Perceived family support	**−0.175 (0.072) ***	−0.036 (0.022)	−0.051 (0.035)
Gender x Sexual orientation	−0.620 (0.808)	−0.552 (0.359)	**−2.460 (0.541) *****
Gender x Age of identification of sexual orientation	0.010 (0.099)	0.027 (0.044)	
Gender x Perceived family support	−0.018 (0.098)		

* *p* < 0.05; ** *p* < 0.01; *** *p* < 0.001. **Bold**: Indicates significance using Bonferroni correction, which adjusted the *p* value to 0.017.

**Table 4 ijerph-19-02427-t004:** Association of internalized sexual stigma with anxiety and depression: Multivariate linear regression analysis.

	Anxiety	Depression
	B (SE)	B (SE)
Gender	**−4.907 (1.074) *****	**−2.857 (0.968) ****
Age	0.115 (0.131)	−0.035 (0.118)
Education	−1.394 (1.239)	−2.202 (1.117)
Sexual orientation	−0.582 (0.849)	−0.787 (0.766)
Social discomfort on the MISS-LG	**0.447 (0.091) *****	**0.306 (0.082) *****
Sexuality on the MISS-LG	**0.438 (0.189) ***	0.198 (0.171)
Identity on the MISS-LG	0.264 (0.122) *	**0.294 (0.110) ****

MISS-LG: Measure of Internalized Sexual Stigma for Lesbians and Gay Men. * *p* < 0.05; ** *p* < 0.01; *** *p* < 0.001. **Bold**: Indicates significance using Bonferroni correction, which adjusted the *p* value to 0.025.

**Table 5 ijerph-19-02427-t005:** Moderating effects of gender on the associations of internalized sexual stigma with anxiety and depression: Multivariate linear regression analysis.

	Anxiety	Depression
	B (SE)	B (SE)
Gender	−0.585 (2.924)	−3.033 (2.249)
Age	0.087 (0.131)	−0.049 (0.119)
Education	−1.136 (1.240)	−2.003 (1.122)
Sexual orientation	−0.478 (0.849)	−0.551 (0.775)
Social discomfort on the MISS-LG	**0.646 (0.120) *****	**0.417 (0.110) *****
Sexuality on the MISS-LG	0.333 (0.315)	0.241 (0.173)
Identity on the MISS-LG	**0.299 (0.123) ***	0.076 (0.163)
Gender x Social discomfort on the MISS-LG	**−0.414 (0.163) ***	−0.241 (0.155)
Gender x Sexuality on the MISS-LG	0.265 (0.383)	
Gender x Identity on the MISS-LG		0.403 (0.218)

MISS-LG: Measure of Internalized Sexual Stigma for Lesbians and Gay Men. * *p* < 0.05; *** *p* < 0.001. **Bold**: Indicates significance using Bonferroni correction, which adjusted the *p* value to 0.025.

## Data Availability

The data will be available upon reasonable request to the corresponding authors.

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
