# Peer review of "Internalized Sexual Stigma among Lesbian, Gay, and Bisexual Individuals in Taiwan: Its Related Factors and Association with Mental Health Problems"

_ijerph, 2022, doi:10.3390/ijerph19042427_

Round 1

Reviewer 1 Report

This cross-sectional study examines the relationships of individual factors and age of identification of sexual and perceived family support with Internalized Sexual Stigma (ISS), and the associations of ISS with mood problems and the moderating effects of gender on the associations among lesbian, gay, and bisexual (LGB) young adults in Taiwan.  Authors assert a need for this study due to the fact that unfavorable attitudes towards LGB in Taiwan are deep-rooted.  Resultantly, LGB individuals may develop internalized stigma, which may compromise their mental health and relationships, lead to increased risky sexual behavior, and decrease intention to access medical care services.  The review is as follows:

  1. Line 20 – In abstract, check spelling in “This quantitate study”.
  2. For the Methods, were participants provided incentives (e.g., rewards) for participation?
  3. Line 124 – Where was the study room located in which participants completed surveys? Were the surveys only completed in person or were they completed in other modes (e.g., online, phone)?  Were participants assured of confidentiality during the informed consent process?
  4. For the Statistical Analyses section, authors should explain the rationale for adjusting the p-values, to aid the lay reader.
  5. Line 23 – In Table 1, the ranges for Perceived family support, Internalized sexual stigma, Social discomfort, Sexuality, Identity, Anxiety, and Depression are unclear. If the items were rated on a 4-point Likert scale, it is not clear what the range numbers (0-76) in the table represent. If the numbers are for range in years, the authors should specify this.
  6. Line 229 – The word “Bald” is unclear. Do authors mean ‘Bold’?
  7. Table 2 is a bit intricate and may be difficult to comprehend by the lay reader.
  8. Lines 331-333 – This sentence is incomplete - “In addition to interventions for LGB individuals, intervention programs to reduce public stigma toward sexual minorities and enhance family support for such individuals, which may help reduce ISS in LGB individuals”.
  9. Authors should consider proposing policy considerations within the Implications section.
  10. Lines 344-345 – Check wording in “Fourth, the question inquiring participants’ gender identity contained the binary classification of man and woman only but no the options of…”.

Overall, this is an interesting and unique paper on an important topic.  Tending to some clarifying questions, including about the methodology, may help to improve the paper.

Author Response

We appreciated your valuable comments. As discussed below, we have revised our manuscript with underlines based on your suggestions. Please let us know if we need to provide anything else regarding this revision.

Comment 1

Line 20 – In abstract, check spelling in “This quantitate study”.

Response

Thank you for your reminding. We corrected it into “This quantitative study”. Please refer to line 20.

Comment 2

For the Methods, were participants provided incentives (e.g., rewards) for participation?

Response

Yes, we provided 500 new Taiwan dollars (about 18 US dollars) for every participant as a reward. We added it into line 131-132.

Comment 3

Line 124 – Where was the study room located in which participants completed surveys? Were the surveys only completed in person or were they completed in other modes (e.g., online, phone)?  Were participants assured of confidentiality during the informed consent process?

Response

In this study, participants completed the research questionnaires in the study rooms of the psychiatry research department affiliated to a university hospital in person and being assured that their responses to the research questionnaire would be confidential. We added it into line 128-131.

Comment 4

For the Statistical Analyses section, authors should explain the rationale for adjusting the p-values, to aid the lay reader.

Response

Thank you for your suggestion. We added “Because of multiple comparisons” into line 190 and 194 to explain the rationale for adjusting the p-values

Comment 5

Line 23 – In Table 1, the ranges for Perceived family support, Internalized sexual stigma, Social discomfort, Sexuality, Identity, Anxiety, and Depression are unclear. If the items were rated on a 4-point Likert scale, it is not clear what the range numbers (0-76) in the table represent. If the numbers are for range in years, the authors should specify this.

Response

Thank you for your reminding.

  • We added the numbers of items measuring Perceived family support (5 items), the Social Discomfort (7 items), Sexuality (5 items) and Identity (5 items) dimensions of Internalized sexual stigma, Anxiety (20 items), and Depression (20 items) into Table 1 to make the ranges of the total scores clear. Please refer to Table 1.
  • Moreover, Depression should be graded on a 4-point scale “from 0 to 3” but not from 1 to 4. We corrected the error. Please refer to line 175-176.
  • We labelled “years” as the units of mean, SD and ranges for age and age of identification of sexual orientation. Please refer to Table 1.

Comment 6

Line 229 – The word “Bald” is unclear. Do authors mean ‘Bold’?

Response

Thank you for your reminding. We meant “Bold” but not “Bald.” We corrected them and please refer to line 220, 232, 243 and 256.

Comment 7

Table 2 is a bit intricate and may be difficult to comprehend by the lay reader.

Response

Thank you for your comment. In the revised manuscript, we made the revisions as below:

  • We split original Table 2 into two new tables accordingly: Table 2 shows the non-gender-moderated regression results and Table 3 shows the gender-moderated results. We also revised the contents of the text: “Table 2 presents the results of the multivariate linear regression analysis of the associations between demographics, sexual orientation factors, and perceived family support and the three ISS” (line 209-211) and “Table 3 presents the results of the multivariate linear regression analysis of gender’s moderating effects on the association between the three ISS dimensions and related factors.” (line 222-223)
  • We split original Table 3 into two new tables accordingly: Table 4 shows the non-gender-moderated regression results and Table 5 shows the gender-moderated results. We also revised the contents of the text: “Table 4 provides the results of the multivariate linear regression analysis of the associations between ISS and anxiety and depression” (line 234-235) and “Table 5 provides the results of the multivariate linear regression analysis of gender’s moderating effects on the associations between ISS and anxiety and depression.” (line 245-246)

Comment 8

Lines 331-333 – This sentence is incomplete - “In addition to interventions for LGB individuals, intervention programs to reduce public stigma toward sexual minorities and enhance family support for such individuals, which may help reduce ISS in LGB individuals”.

Response

Thank you for your reminding. We rewrote the content of the Implication section as below. Please refer to line 327-344.

The findings of the present study highlight the value of developing strategies for the prevention of ISS in LGB individuals. While acceptance of LGB individuals has substantially improved across the past several decades, total acceptance is far from complete due to the continued prevalence of generally unwarranted negative familial and cultural stereotypes and thus further efforts to modify outdated beliefs with regard to LGB individuals are warranted. Constitutions, laws and anti-discrimination policies at the national level that include protection from discrimination on the grounds of sexual orientation is of fundamental importance [76]. Broadening understanding of LGB culture and awareness of prejudices toward LGB individuals in educational settings, workplaces, and home environments are also the necessary steps to help reduce sexuality-related stigma [76,77]. Public health strategies addressing attitudes to sexual orientation and promoting the changes of attitudes toward sexual minority among the general population may contribute to diverse affirmative cultural scripts regarding LGB individuals’ lives [76,78]. Although several studies have developed the programs for reducing ISS among gay and bisexual men [79-81], there is no study directly examining the effects of interventions addressing familial and cultural stigma on ISS among LGB individuals. Meanwhile, given the effect of gender on the associations between ISS and sexual orientation and anxiety identified in this study, intervention programs should be gender-specific.

Comment 9

Authors should consider proposing policy considerations within the Implications section.

Response

We added policy considerations into the Implication section as described in Response to Comment 8. Please refer to line 327-344.

Comment 10

Lines 344-345 – Check wording in “Fourth, the question inquiring participants’ gender identity contained the binary classification of man and woman only but no the options of…”.

Response

Thank you for your suggestion. We revised this sentence into “Fourth, the options for the question inquiring participants’ gender identity contained man and woman only but no other gender identities such as transgender, gender nonbinary, and genderqueer.” Please refer to line XXX.

Comment 11

Overall, this is an interesting and unique paper on an important topic. Tending to some clarifying questions, including about the methodology, may help to improve the paper.

Response

Thank you for your comments on our manuscript. They are helpful to improve the manuscript.

Reviewer 2 Report

This is an interesting and, for the most part, a well-written, informative report of the results of a survey of 1000 LGB individuals regarding the association of various demographic and psychiatric factors with internalized sexual stigma. It will certainly be of interest to the readers of Environmental Research and Public Health.  Not only is it a nice piece of work from a methodological standpoint, but it addresses a research area that has generally been neglected within the literature.  Once a few issues are addressed, the paper should be published as soon as conveniently possible.

  1. Although the English in this manuscript is quite good, it nonetheless contains several minor errors that should be corrected. Examples can be found in Line 27, where the first “and” should be deleted; Line 34, which should read “as well as on the association between…”; Line 35, which should read “should be routinely assessed by mental health…”; Line 36, where “unit” is misspelled; and Line 116, in which “into study” should be deleted since it is redundant in this context.
  2. Please define “structural stigma” on line 54.
  3. Does the example on line 68 explicitly mean: As a gay man I feel awkward when I have sex with a woman? Or does it mean, As a lesbian I feel awkward when I have sex with a woman? Please clarify.
  4. Consider rephrasing line 83 to: “…can highlight factors that should receive emphasis in efforts to better educate the broader society regarding their views on LGB individuals as well as within the context of psychotherapy in the event that an LGB individual feels the need for such therapy.”
  5. Please elaborate on the statement in line 95 that “ISS is positively associated with older age groups in terms of accepting sexual orientation among young LGB individuals.” I’m not clear on exactly what this means.
  6. Please explain more thoroughly what is meant by “in a study room” as stated on line 124. Is this a college classroom? A high school auditorium? An LGB support center?
  7. On line 134, please start the sentence with “Some of the sample items for each dimension…”
  8. For consistency’s sake, please start the paragraph at line 132 with “We used the Chinese version of the MISS-LG…” similar to the opening sentence in paragraphs 2.2.2, 2.2.4, and 2.2.5.
  9. On line 211, correct the typo so the phrase reads “… was identified as being significantly associated with…”
  10. I found Table 2 to be somewhat confusing as presently formatted. First, I believe the first predictor that is currently labelled “Male” should be labelled “Gender,” the third predictor should be labelled “Education,” and the fourth should be labelled “Sexual Orientation.” The text can then explain for instance that the significance on the gender factor was because males experienced more social discomfort, greater problems with being identified as non-heterosexual, and more pessimism over the quality and endurance of same-sex relationships than females. Secondly, I think it would be better to split the non-gender-moderated regression results and the gender-moderated results into 2 separate tables rather than referring to Models 1,3, and 5 versus Models 2, 4, and 6 within the same table. Third, (this suggestion may be one that would be handled by the journal editor), the underscore under the initial heading row should not be continued across the 3 scales.
  11. Please also revise the current Table 3 so that it is consistent with the recommendations made above (for Table 2).
  12. My final recommendation is that the authors reconsider the implications of the present research in paragraph 4.3 (and in the Conclusions section). Presently, the authors seem to be “blaming the victims” for the problem of ISS as well as suggesting that people with ISS are generally mentally ill. Stating that the present results indicate the need for “prevention” and “detection” of ISS suggests that ISS is some sort of psychiatric disease that inevitably exists throughout the LGB community and inevitably leads to treatment-essential mental illness. While it is certainly the case that ISS is a stressor, I don’t believe the present research proves that ISS is a profound driver of psychotherapy throughout the LGB community.  Wouldn’t it be better to focus on the need for family education and cultural change in order to reduce the amount of ISS in otherwise healthy individuals or to mitigate the potential impact of familial and societal misconceptions that might ultimately contribute to poor psychological adjustment? My recommendation is that the primary implications of the present research should be the ones listed in lines 331 and 333 and not the ones listed in lines 320 and 321.  I recommend that the authors be careful to avoid couching their findings in a manner that supports the existing societal prejudice against LGB persons by suggesting that gay and bisexual people are likely to be mentally ill.  Note that the Conclusions paragraph also implies that most or all LGB are going to end up in psychotherapy since the results of the current research are considered most important to “mental health professionals” rather than being most important to the families and social institutions within Asian (and most other) societies.  I’m rather uncomfortable with this emphasis on mental illness and mental health intervention. Instead, it would be more beneficial to the LGB community if the most important implication of the present research was that “while acceptance of gay and lesbian and bisexual people has substantially improved across the past several decades, total acceptance is far from complete due to the continued prevalence of generally unwarranted negative familial and cultural stereotypes, and thus further efforts to modify outdated beliefs with regard to LGB persons are warranted.” Please carefully consider the “take-home message” of the present findings.

Once the above issues have been addressed, the paper should be published.

Author Response

We appreciated your valuable comments. As discussed below, we have revised our manuscript with underlines based on your suggestions. Please let us know if we need to provide anything else regarding this revision.

Comment 1

Although the English in this manuscript is quite good, it nonetheless contains several minor errors that should be corrected. Examples can be found in:

  • Line 27, where the first “and” should be deleted;
  • Line 34, which should read “as well as on the association between…”;
  • Line 35, which should read “should be routinely assessed by mental health…” and Line 36, where “unit” is misspelled;
  • Line 116, in which “into study” should be deleted since it is redundant in this context.

Response

Thank you for your reminding. We corrected them, including:

  • Deleting the first “and” from line 27.
  • Revising the sentence into “Gender had moderating effects on the association between the identity dimension of ISS and sexual orientation as well as between the social discomfort dimension of ISS and anxiety” in line 32-34.
  • Revising the sentence into “…assessed by mental health units” in line 35.
  • Deleting “into study” from line 117.
  • We also corrected other typos and errors in the manuscript, for example, “quantitative” in line 20; “Bold” in line 220, 232, 243 and 256; “from 0…3” in line 175-176; deleting “suicidality” from Table 4.

Comment 2

Please define “structural stigma” on line 54.

Response

We added the definition for structural stigma as below. Please refer to line 53-54.

“…discrimination and stigmatization at the institutional and sociocultural levels e.g., the ban on same-sex marriage…

Comment 3

Does the example on line 68 explicitly mean: As a gay man I feel awkward when I have sex with a woman? Or does it mean, As a lesbian I feel awkward when I have sex with a woman? Please clarify.

Response

Thank you for your suggestion. In order to reduce confusion, we used another item (“Gay men can only have flings/one-night stands”) to replace the original example (“When I have sex with a woman, I feel awkward”) for illustrating the sexuality dimension of the MISS-LG. Please refer to line 68.

Comment 4

Consider rephrasing line 83 to: “…can highlight factors that should receive emphasis in efforts to better educate the broader society regarding their views on LGB individuals as well as within the context of psychotherapy in the event that an LGB individual feels the need for such therapy.”

Response

Thank you for your suggestion. It is really helpful. We revised this sentence accordingly. Please refer to line 82-85.

Comment 5

Please elaborate on the statement in line 95 that “ISS is positively associated with older age groups in terms of accepting sexual orientation among young LGB individuals.” I’m not clear on exactly what this means.

Response

We revised this sentence into “ISS is positively associated with delayed acceptance of sexual orientation among young LGB individuals.” Please refer to line 96-98.

Comment 6

Please explain more thoroughly what is meant by “in a study room” as stated on line 124. Is this a college classroom? A high school auditorium? An LGB support center?

Response

Thank you for your comment. We revised it into “in the study rooms of the psychiatry research department affiliated to a university hospital”. Please refer to line 129.

Comment 7

On line 134, please start the sentence with “Some of the sample items for each dimension…”

Response

We revised it accordingly. Please refer to line 139-140.

Comment 8

For consistency’s sake, please start the paragraph at line 132 with “We used the Chinese version of the MISS-LG…” similar to the opening sentence in paragraphs 2.2.2, 2.2.4, and 2.2.5.

Response

Thank you for your suggestion. We revised them accordingly, including:

2.2.1. “We used the 17-item Chinese version [42] of the MISS-LG [14]…” (line 137)

2.2.2 “We used the 5-item traditional Chinese version [43] of Family APGAR Index [44]…” (line 151)

2.2.4 “We used the 20-item traditional Chinese version [46] of the self-administered State-Trait Anxiety Inventory-State subscale [47]…” (line 163-164)

2.2.5 “We used the 20-item traditional Chinese version [49] of the self-administered Center for Epidemiological Studies Depression Scale [50]…” (line 172-173)

Comment 9

On line 211, correct the typo so the phrase reads “… was identified as being significantly associated with…”

Response

Thank you for your reminding. We corrected it accordingly. Please refer to line 215.

Comment 10

I found Table 2 to be somewhat confusing as presently formatted.

Comment 10-1

First, I believe the first predictor that is currently labelled “Male” should be labelled “Gender,” the third predictor should be labelled “Education,” and the fourth should be labelled “Sexual Orientation.” The text can then explain for instance that the significance on the gender factor was because males experienced more social discomfort, greater problems with being identified as non-heterosexual, and more pessimism over the quality and endurance of same-sex relationships than females.

Response

Thank you for your suggestion. We changed them into “Male,” “Education,” and the “Sexual Orientation.” Please refer to Tables 2 to 5.

Comment 10-2

Secondly, I think it would be better to split the non-gender-moderated regression results and the gender-moderated results into 2 separate tables rather than referring to Models 1,3, and 5 versus Models 2, 4, and 6 within the same table.

Response

We split original Table 2 into two new tables accordingly: Table 2 shows the non-gender-moderated regression results and Table 3 shows the gender-moderated results. We also revised the contents of the text: “Table 2 presents the results of the multivariate linear regression analysis of the associations between demographics, sexual orientation factors, and perceived family support and the three ISS dimensions.” (line 209-211) and “Table 3 presents the results of the multivariate linear regression analysis of gender’s moderating effects on the association between the three ISS dimensions and related factors.” (line 222-223)

Comment 10-3

Third, (this suggestion may be one that would be handled by the journal editor), the underscore under the initial heading row should not be continued across the 3 scales.

Response

We modified Tables 2 to 5 to discontinued the underscore under the initial heading row across the 3 scales.

Comment 11

Please also revise the current Table 3 so that it is consistent with the recommendations made above (for Table 2).

Response

Thank you for your suggestion. In addition to make the revisions as the responses to 10-1 and 10-3, we split original Table 3 into two new tables accordingly: Table 4 shows the non-gender-moderated regression results and Table 5 shows the gender-moderated results. We also revised the contents of the text: “Table 4 provides the results of the multivariate linear regression analysis of the associations between ISS and anxiety and depression” (line 234-235) and “Table 5 provides the results of the multivariate linear regression analysis of gender’s moderating effects on the associations between ISS and anxiety and depression.” (line 245-246)

Comment 12

My final recommendation is that the authors reconsider the implications of the present research in paragraph 4.3 (and in the Conclusions section). Presently, the authors seem to be “blaming the victims” for the problem of ISS as well as suggesting that people with ISS are generally mentally ill. Stating that the present results indicate the need for “prevention” and “detection” of ISS suggests that ISS is some sort of psychiatric disease that inevitably exists throughout the LGB community and inevitably leads to treatment-essential mental illness. While it is certainly the case that ISS is a stressor, I don’t believe the present research proves that ISS is a profound driver of psychotherapy throughout the LGB community. Wouldn’t it be better to focus on the need for family education and cultural change in order to reduce the amount of ISS in otherwise healthy individuals or to mitigate the potential impact of familial and societal misconceptions that might ultimately contribute to poor psychological adjustment? My recommendation is that the primary implications of the present research should be the ones listed in lines 331 and 333 and not the ones listed in lines 320 and 321. I recommend that the authors be careful to avoid couching their findings in a manner that supports the existing societal prejudice against LGB persons by suggesting that gay and bisexual people are likely to be mentally ill.  Note that the Conclusions paragraph also implies that most or all LGB are going to end up in psychotherapy since the results of the current research are considered most important to “mental health professionals” rather than being most important to the families and social institutions within Asian (and most other) societies.  I’m rather uncomfortable with this emphasis on mental illness and mental health intervention. Instead, it would be more beneficial to the LGB community if the most important implication of the present research was that “while acceptance of gay and lesbian and bisexual people has substantially improved across the past several decades, total acceptance is far from complete due to the continued prevalence of generally unwarranted negative familial and cultural stereotypes, and thus further efforts to modify outdated beliefs with regard to LGB persons are warranted.” Please carefully consider the “take-home message” of the present findings.

Response

We appreciate your valuable comment and suggestion. Accordingly, we rewrote the content of the Implication and Conclusion section as below and focused on how to reduce ISS by advocating sexuality equality.

4.3. Implications

The findings of the present study highlight the value of developing strategies for the prevention of ISS in LGB individuals. While acceptance of LGB individuals has substantially improved across the past several decades, total acceptance is far from complete due to the continued prevalence of generally unwarranted negative familial and cultural stereotypes and thus further efforts to modify outdated beliefs with regard to LGB individuals are warranted. Constitutions, laws and anti-discrimination policies at the national level that include protection from discrimination on the grounds of sexual orientation is of fundamental importance [76]. Broadening understanding of LGB culture and awareness of prejudices toward LGB individuals in educational settings, workplaces, and home environments are also the necessary steps to help reduce sexuality-related stigma [76,77]. Public health strategies addressing attitudes to sexual orientation and promoting the changes of attitudes toward sexual minority among the general population may contribute to diverse affirmative cultural scripts regarding LGB individuals’ lives [76,78]. Although several studies have developed the programs for reducing ISS among gay and bisexual men [79-81], there is no study directly examining the effects of interventions addressing familial and cultural stigma on ISS among LGB individuals. Meanwhile, given the effect of gender on the associations between ISS and sexual orientation and anxiety identified in this study, intervention programs should be gender-specific.” Please refer to line 327-344.

  1. Conclusion
    Further efforts to modify the public’s prejudice with regard to LGB individuals are warranted. Family education and cultural change aiming to reduce public stigma against LGB individuals are needed to mitigate ISS among LGB individuals.” Please refer to line 361-364.

Comment 13

Once the above issues have been addressed, the paper should be published.

Response

Thank you for your comments on our manuscript. They are helpful to improve the manuscript.

Round 2

Reviewer 1 Report

The authors have done well to address the requested feedback.  The manuscript is clearer and improved, including the expanded discussion on policy considerations.  The only thing that is still not clear is Table 1 and the Internalized sexual stigma on the MISS-LG, anxiety, and depression score.  It is not clear what is considered low-to-high for the mean scores.  Otherwise, authors have sufficiently addressed the requested feedback.

Author Response

We appreciated your valuable comment. As discussed below, we have revised our manuscript with underlines based on your suggestions. Please let us know if we need to provide anything else regarding this revision.

Comment

The only thing that is still not clear is Table 1 and the Internalized sexual stigma on the MISS-LG, anxiety, and depression score. It is not clear what is considered low-to-high for the mean scores.

Response

Thank you for your comment. In the revised manuscript, we made the revisions to make the meanings of mean scores in Table 1 clear:

  • We transformed the total scores for the three dimensions of the MISS-LG into the scores for each item:

After transforming, the mean score for each item on the dimension of social discomfort (2.4) was the highest, followed by the scores for identity (2.0) and sexuality (1.8).” Please refer to line 212-214.

  • We used the cutoff score of 40 commonly used for screening anxiety disorder to present the meaning of STAI-S mean score:

The score of 40 is commonly used to define probable clinical levels of anxiety [49].” Please refer to line 174-175.

The mean score (SD) for the severity of anxiety was 40.8 (12.7) and slightly higher than the cutoff 40 that was commonly used for screening anxiety disorders [49].” Please refer to line 214-216.

  • We used the cutoff score of 16 commonly used for screening major depressive disorder to present the meaning of CES-D mean score:

The score of 16 is commonly used for screening major depressive disorder in the general population [54].” Please refer to line 185-186.

The mean score (SD) for depression was 18.8 (11.2) and higher than the cutoff 16 that was commonly used for screening major depressive disorders [54].” Please refer to line 216-218.

This manuscript is a resubmission of an earlier submission. The following is a list of the peer review reports and author responses from that submission.